# Dynamics Analysis of Penetration against Pebble-Concrete Target

**DOI:** 10.3390/ma15051675

**Published:** 2022-02-23

**Authors:** Huan Yan, Lei Jin, Shiqiao Gao, Xiao Xu, Zezhang Li, He Fu

**Affiliations:** 1State Key Laboratory of Explosion Science and Technology, Beijing Institute of Technology, Beijing 100081, China; 3120170104@bit.edu.cn (H.Y.); xuxiao_1990@126.com (X.X.); fuhekkxx@outlook.com (H.F.); 2Beijing Institute of Automatic Control Equipment, Beijing 100074, China; zezhang_li@163.com

**Keywords:** couple stress theory, pebble-concrete target, normal cavity expansion, penetration, high-speed impact

## Abstract

The classical continuum mechanics theory cannot sufficiently describe the effect of pebbles on projectile, which leads to a large calculation error. In this paper, an orthogonal curvilinear coordinate system is constructed, which effectively describes and perfects the normal cavity expansion theory. A couple stress theory based on the normal cavity expansion is proposed in which not only the tangential movements but also the rotations of the concrete medium are considered. According to the high-speed impact of pebble concrete, combined with dynamic equations and the FE simulation, the theoretical and simulation results of pebble particles scale on warhead resistance are compared. It is shown that, the larger the scale of pebble particles, the stronger the effect of rotation on the resistant force applied on the warhead.

## 1. Introduction

In earlier research on the penetration resistance characteristics of concrete target, several explorations were carried out on many aspects, such as penetration depth and projectile mass loss, and research results such as semi-empirical formulas and projectile mass loss characterization models accumulated. However, most of these studies are based on macroscopic models of concrete material. With further research of penetrating concrete, the FE simulation of the 3D meso-scale model has gradually assisted new technologies, such as mapping technology and deep learning [1,2]. In recent years, Liu et al. [3,4] have obtained the penetration mechanism of high-speed penetration of reinforced concrete using a 3D meso-scale model. Lv [2] reproduced the formation process of typical failure patterns of concrete specimens by using this model. Zhang et al. [5] focused research on the penetration resistance of concrete based on the meso-scale model. In order to enhance the strength of the structure, a number of stones, such as pebbles, are usually added to the concrete, and sometimes the size of the stones is relatively large. In the experiments, as shown in Figure 1a,b, ellipsoidal pebbles can be seen in both crater and collapsed waste, and Figure 1c shows the warhead surface condition after the penetration experiment [6]. The warhead has no obvious deformation, but these pebbles caused varying degrees of scratches on the surface of the warhead. In the classical continuum mechanics theory, only the translational movement of the particles is considered, which does not adequately describe the motion of the pebbles in space. When the size of the pebbles is relatively small compared with the size of the projectile, it is possible to ignore the influence of its size and regard it as a three-dimensional translational mass point. However, when the size of the pebbles is too large to be negligible compared with the projectile, if it is still considered as a particle, it will cause a large calculation error. This is because the actual pebbles will rotate in addition to the translation of the center of mass during the projectile penetration.

For the research of projectiles penetrating against concrete targets, the classic analytical method is based on the cavity expansion theory, including the cylindrical expansion theory and the spherical cavity expansion theory [7,8,9,10,11,12,13,14]. Among them, the theory of cylindrical cavity expansion believes that the target medium only moves in the direction of the radius of the projectile shrank during penetration, thus forming radially expanding cavities. The theory of spherical cavity expansion believes that the target medium moves in the normal direction of corresponding inward ball of the projectile surface, thus forming a cavity consisting of a series of balls enveloped by the surface of the warhead. In order to better adapt to the shape of general warheads, Gao et al. [15,16,17,18] proposed the normal expansion theory, which overcomes the limitation of these two theories and develops them. In this normal expansion theory, it is believed that the movement of the medium during the expansion of the cavity is along the outer normal direction of the surface of the warhead, so it is more adaptable to the situation of different warhead shapes. The cavity expansion theory does not consider the rotation of the target medium and the translation along the tangent direction of the warhead surface. For general fine-grained concrete media, since the tangential motion of the media only occurs in a very narrow area, traditional methods are effective to analyze and solve the problem of normal pressure on the warhead surface. For the concrete media containing large particles, such as pebbles, the tangential force caused by the viscous friction will not only generate translational movements of pebble particles along the tangential direction of warhead surface, but also cause the pebble particles to rotate. At the same time, the range of tangential motion of the medium will also be very wide. Therefore, the traditional theory of the cavity expansion, which only considers the normal motion of the medium, can no longer solve the problem of normal pressure on the warhead surface effectively.

In order to analyze the penetration of the concrete media containing pebbles effectively, this paper proposes the continuum mechanics theory of the couple stress [19,20,21,22]. It considers both additional tangential translational motion and additional rotation of media particles based on the theory of normal cavity expansion. As shown in Figure 2a, the target medium is divided into two regions, which are the response region and the free region, and the interface of these two regions is the wave front of the shock expansion wave. The wave front propagates along the normal direction of warhead surface at a wave velocity. It can be seen in Figure 2b that the media particles of the response region have translational motion and rotational motion. The translational motion not only has a component along the normal direction of the cavity, but also has a component along the tangential direction of the cavity. The theoretical analysis is further verified by paying attention to the deceleration and velocity through the FE simulation and experiment. Finally, the correctness of the theoretical analysis is further verified by comparing the theoretical calculation with the FE simulation.

## 2. The Basis of the Couple Stress Theory

### 2.1. The Orthogonal Curvilinear Coordinate System

To facilitate analysis, an orthogonal curvilinear coordinate system (x,θ,φ) at a certain point of the warhead surface is shown in Figure 3, where x is the normal coordinate of the warhead surface, θ is the circumferential coordinate, φ is the meridian coordinate, $A^{\prime }$ is a point on the warhead surface, A is a point of the concrete target medium, R is the curvature radius of the warhead surface, $\bar{R}$ is a part of curvature radius cut by $x_{3}$ axis, $e_{x}$, $e_{\theta }$ and $e_{\varphi }$ are orthogonal unit basis vectors in the directions x, θ and φ.

The gradient operator vector is represented as
(1)$$\nabla =\mathrm{grad}=\left(\frac{\partial }{\partial x},\frac{1}{\left(\bar{R}+x\right)\sin\varphi }\frac{\partial }{\partial \theta },\frac{1}{R+x}\frac{\partial }{\partial \varphi }\right)$$

The components of substantive derivative of velocity $v=\left(v_{x},v_{\theta },v_{\varphi }\right)$ are written by
(2)$$\{\begin{array}{l} {\left(\frac{\mathrm{Dv}}{\mathrm{Dt}}\right)}_{x}=\frac{{\mathrm{Dv}}_{x}}{\mathrm{Dt}}-\frac{v_{\varphi }{}^{2}}{R+x}-\frac{v_{\theta }{}^{2}}{\bar{R}+x}\\ {\left(\frac{\mathrm{Dv}}{\mathrm{Dt}}\right)}_{\theta }=\frac{{\mathrm{Dv}}_{\theta }}{\mathrm{Dt}}+\frac{v_{x}v_{\theta }}{R+x}+\frac{v_{\varphi }v_{\theta }\cot\varphi }{\bar{R}+x}\\ {\left(\frac{\mathrm{Dv}}{\mathrm{Dt}}\right)}_{\varphi }=\frac{{\mathrm{Dv}}_{\varphi }}{\mathrm{Dt}}+\frac{v_{x}v_{\varphi }}{R+x}-\frac{v_{\theta }{}^{2}\cot\varphi }{\bar{R}+x} \end{array}$$

The curl of velocity $v=\left(v_{x},v_{\theta },v_{\varphi }\right)$ is
(3)$$\nabla \times v=\frac{1}{\left(R+x\right)\left(\bar{R}+x\right)\sin\varphi }\left|\begin{array}{ccc} e_{x} & \left(\bar{R}+x\right)\sin{\varphi e}_{\theta } & \left(R+x\right)e_{\varphi }\\ \frac{\partial }{\partial x} & \frac{\partial }{\partial \theta } & \frac{\partial }{\partial \varphi }\\ v_{x} & \left(\bar{R}+x\right)\sin{\varphi v}_{\theta } & \left(R+x\right)v_{\varphi } \end{array}\right|$$

From this, the angular velocity can be obtained
(4)$${\dot{\omega }}_{\theta }=\frac{1}{2}{\left(\nabla \times v\right)}_{\theta }=\frac{1}{2}\left[\frac{\partial v_{\varphi }}{\partial x}+\frac{v_{\varphi }}{\left(R+x\right)}-\frac{1}{\left(R+x\right)}\frac{\partial v_{x}}{\partial \varphi }\right]$$

The divergency of velocity $v=\left(v_{x},v_{\theta },v_{\varphi }\right)$ is
(5)$$\mathrm{div}v=\frac{\partial v_{x}}{\partial x}+\frac{v_{x}}{R+x}+\frac{v_{x}}{\bar{R}+x}+\frac{1}{R+x}\frac{\partial v_{\varphi }}{\partial \varphi }+\frac{v_{\varphi }\cot\varphi }{\bar{R}+x}+\frac{1}{\left(\bar{R}+x\right)\sin\varphi }\frac{\partial v_{\theta }}{\partial \theta }$$

Stress can be represented by a second-order tensor A, and the divergence of stress is a vector. Its three components can be represented by the formula $\left(\mathrm{div}A{)}_{i}=\mathrm{div}(A\cdot e_{i}\right)-\mathrm{tr}\left[A\cdot \left(\nabla e_{i}\right)\right].$ The components of the second-order tensor A are as follows:(6)$${\left(\mathrm{div}A\right)}_{x}=\frac{\partial A_{\mathrm{xx}}}{\partial x}+\frac{1}{R+x}\frac{\partial A_{\varphi x}}{\partial \varphi }+\frac{1}{\left(\bar{R}+x\right)\sin\varphi }\frac{\partial A_{\theta x}}{\partial \theta }+\frac{1}{R+x}\left(A_{\mathrm{xx}}-A_{\varphi \varphi }\right)\;+\frac{1}{\bar{R}+x}\left(A_{\mathrm{xx}}-A_{\theta \theta }+A_{\varphi x}\cot\varphi \right)$$
(7)$${\left(\mathrm{div}A\right)}_{\theta }=\frac{\partial A_{x\theta }}{\partial x}+\frac{1}{R+x}\frac{\partial A_{\varphi \theta }}{\partial \varphi }+\frac{1}{\left(\bar{R}+x\right)\sin\varphi }\frac{\partial A_{\theta \theta }}{\partial \theta }+\frac{1}{R+x}\left(A_{x\theta }+A_{\theta x}\right)\;+\frac{1}{\bar{R}+x}\left[A_{x\theta }+\left(A_{\theta \varphi }+A_{\varphi \theta }\right)\cot\varphi \right]$$
(8)$${\left(\mathrm{div}A\right)}_{\varphi }=\frac{\partial A_{x\varphi }}{\partial x}+\frac{1}{R+x}\frac{\partial A_{\varphi \varphi }}{\partial \varphi }+\frac{1}{\left(\bar{R}+x\right)\sin\varphi }\frac{\partial A_{\theta \varphi }}{\partial \theta }+\frac{1}{R+x}\left(A_{x\varphi }+A_{\varphi x}\right)\;+\frac{1}{\bar{R}+x}\left[A_{x\varphi }+\left(A_{\varphi \varphi }-A_{\theta \theta }\right)\cot\varphi \right]$$
where $A_{\mathrm{ij}}$ are components of the second-order tensor A.

### 2.2. Basic Equations

When only considering the rotation with the material, it belongs to the couple stress theory, which belongs to the category of modern continuous medium mechanics. According to the couple stress theory, for the adiabatic process, the basic equations of the medium in the response region during dynamic deformation include mass conservation equation, momentum conservation equation and angular momentum conservation equation, which can be written as
(9)$$\frac{\partial \rho }{\partial t}+\mathrm{div}\left(\rho v\right)=0$$
(10)$$\rho \frac{\mathrm{Dv}}{\mathrm{Dt}}=\mathrm{div}\sigma^{\prime }+\rho b$$
(11)$$\rho_{s}I\cdot \ddot{\omega }={\rho m}_{v}+\mathrm{divm}+\in :\sigma^{\prime }$$
where ρ is the average mass density of the medium (material), $\rho_{s}$ is the mass density of pebbles, v is the particle velocity of the material, $\sigma^{\prime }$ is the stress, b is the body force, m is the couple stress, $m_{v}$ is the body couple stress and $\dot{\omega }$ is the angular velocity, I is the rotational inertia per unit mass of the stone particles, ∈ is a symbol tensor whose components are expressed by $e_{\mathrm{ijk}}$.

For the high-speed impact, the medium response region is continuously expanding. Such an expansion represents the propagation of shock waves. There is a shock wave front between the response region and the free region. The shock wave front is a discontinuity surface. The stress $\sigma^{\prime }$, the couple stress m, the density ρ, the particle velocity v and the angular velocity $\dot{\omega }$ on both sides of the discontinuity surface are discontinuous. According to the theory of discontinuity surface and the principles of mass conservation, momentum conservation and angular momentum conservation on both sides of the discontinuity surface, the following relationship can be obtained:(12)Δ(ρυ)=0
(13)$$\Delta \left(\rho \upsilon v\right)+e_{n}\cdot \Delta \sigma =0$$
(14)$$\Delta \left(\rho_{s}\upsilon I\cdot \dot{\omega }\right)+e_{n}\cdot \Delta (m+l\times \sigma^{\prime })=0$$
where Δ represents the difference of physical quantity (tensor) between the ahead and behand of the shock wave front, · represents the dot product of the two vectors, $\upsilon =c-v_{n}$, $e_{n}$ is the unit vector in the normal direction of the shock wave front, c is the shock wave velocity and $v_{n}$ is the particle velocity in the normal direction of the shock wave front.

## 3. The Couple Stress Model Based on Normal Cavity Expansion

### 3.1. Dynamic Equations of Medium in the Response Region

To establish the couple stress model of continuum mechanics based on the normal cavity expansion, the following assumptions are made: (1) the shock expansion wave expands in the normal direction of warhead surface at the wave velocity $c_{x}$, (2) the velocity of material particles in the response region includes the component along the normal direction and that along the tangent direction of the cavity, (3) there is a asymmetric stress $p_{\varphi }$ caused by the viscous force in the meridian direction of the warhead surface, (4) there is a rotational motion of the material particles caused by the viscous force in the circumferential direction and (5) the rotations in the other two directions can be ignored. According to these assumptions, and considering the axial symmetry in the circumferential direction, the velocity vector can be described as $v=\left(v_{x},0,v_{\varphi }\right)$ and the angular velocity vector can be described as $\dot{\omega }=\left(0,{\dot{\omega }}_{\theta },0\right)$. There are the following equations:(15)$$v=v_{x}e_{x}+v_{\varphi }e_{\varphi }$$
(16)$$\dot{\omega }={\dot{\omega }}_{\theta }e_{\theta }$$
(17)$$c=c_{x}e_{x}$$
(18)$$\sigma^{\prime }=\left[\begin{array}{ccc} -p_{x} & 0 & p_{\varphi }\\ 0 & 0 & 0\\ 0 & 0 & 0 \end{array}\right]=\left[\begin{array}{ccc} -p_{x} & 0 & \frac{1}{2}p_{\varphi }\\ 0 & 0 & 0\\ \frac{1}{2}p_{\varphi } & 0 & 0 \end{array}\right]+\left[\begin{array}{ccc} 0 & 0 & \frac{1}{2}p_{\varphi }\\ 0 & 0 & 0\\ \frac{-1}{2}p_{\varphi } & 0 & 0 \end{array}\right]$$
(19)$${\dot{\omega }}_{\theta }=\frac{1}{2}$$
where c is the velocity vector of the shock wave, $p_{x}$ is the normal pressure and $p_{\varphi }$ is the tangential shear stress.

In this paper, the rotation is mainly caused by antisymmetric stress, so the effects of body force, the body couple stress and the couple stress can be ignored. Then, the basic equations of mass conservation equation, momentum conservation equation and angular momentum conservation equation can be rewritten as the following:(20)$$\frac{\partial \rho }{\partial t}+\frac{\partial \left({\rho v}_{x}\right)}{\partial x}+\frac{{\rho v}_{x}}{R+x}+\frac{{\rho v}_{x}}{\bar{R}+x}+\frac{1}{R+x}\frac{\partial \left({\rho v}_{\varphi }\right)}{\partial \varphi }+\frac{{\rho v}_{\varphi }\cot\varphi }{\bar{R}+x}=0$$
(21)$$\{\begin{array}{l} \rho \frac{{\mathrm{Dv}}_{x}}{\mathrm{Dt}}-\rho \frac{v_{\varphi }{}^{2}}{R+x}=-\left(\frac{\partial p_{x}}{\partial x}+\frac{p_{x}}{R+x}+\frac{p_{x}}{\bar{R}+x}\right)\\ \rho \frac{{\mathrm{Dv}}_{\varphi }}{\mathrm{Dt}}+\rho \frac{v_{x}v_{\varphi }}{R+x}=\left(\frac{\partial p_{\varphi }}{\partial x}+\frac{p_{\varphi }}{R+x}+\frac{p_{\varphi }}{\bar{R}+x}\right) \end{array}$$
(22)$$\rho_{s}I_{\theta }\cdot \frac{1}{2}\left[\frac{\partial {\dot{v}}_{\varphi }}{\partial x}+\frac{{\dot{v}}_{\varphi }}{\left(R+x\right)}-\frac{1}{\left(R+x\right)}\frac{\partial {\dot{v}}_{x}}{\partial \varphi }\right]=p_{\varphi }$$
where $I_{\theta }=2l^{2}$ is the rotational inertia of the stone particles per unit mass in the θ direction, and l is the characteristic size of the stone particles.

### 3.2. Boundary Conditions of the Medium in the Response Region

The medium in the response region has two boundaries, one is the boundary (x=0) in contact with the warhead surface, and the other is the boundary interface (x=L) between the response region and the free region.

On the shock wave front, the step formula on both sides of the discontinuity surface can be written by
(23)$$\left(\rho_{\mathrm{sf}}-\rho_{0}\right)c_{x}-\rho_{\mathrm{sf}}v_{x}^{L}=0$$
(24)$$\rho_{\mathrm{sf}}\left(c_{x}-v_{x}^{L}\right)v_{x}^{L}=p_{x}^{L}$$
(25)$$\rho_{\mathrm{sf}}\left(c_{x}-v_{x}^{L}\right)v_{\varphi }^{L}=p_{\varphi }^{L}$$
(26)$$\rho_{s}\left(c_{x}-v_{x}^{L}\right)I_{\theta }{\dot{\omega }}_{\theta }^{L}={\mathrm{lp}}_{\varphi }^{L}$$
where $\rho_{\mathrm{sf}}$ is the density of the compressed medium near the shock wave front, $\rho_{0}$ is the initial density of the material, $v_{x}^{L}$, $v_{\varphi }^{L}$, $p_{x}^{L}$, ${\dot{\omega }}_{\theta }^{L}$ and $p_{\varphi }^{L}$ are the particle normal velocity, tangential velocity, angular velocity, normal stress and tangential stress of the response medium at the boundary (x=L) of the shock wave front, respectively.

At the boundary (x=0) of the contact surface between the warhead and the target, the normal velocity of the material particles $v_{x0}$ is the same as the normal velocity of the warhead surface $v_{\mathrm{xp}}$, which can be expressed by
(27)$$v_{x0}=v_{\mathrm{xp}}$$

Affected by frictional viscous force, there is a proportional relationship between the tangential velocity of the material particles $v_{\varphi 0}$ and the tangential velocity of the warhead surface $v_{\varphi p}$, which can be expressed as
(28)$$v_{\varphi 0}=\beta_{\mathrm{tt}}v_{\varphi p}$$
where $0\leq \beta_{\mathrm{tt}}\leq 1$ is the adhesion coefficient on the interface between the projectile and the target. The larger the value, the stronger the adhesion between the projectile and the target.

According to the geometric relationship of the warhead surface, the relationships between the normal velocity $v_{\mathrm{xp}}$, the tangential velocity $v_{\varphi p}$ and the axial penetration velocity $v_{z}$ of the projectile at a certain point on the warhead surface are as follows:(29)$$v_{\mathrm{xp}}=v_{z}\cos\varphi$$
(30)$$v_{\varphi p}=-v_{z}\sin\varphi$$
where $v_{z}$ is the axial penetration velocity of the projectile.

Thus, at a certain point on the warhead surface, there is the following velocity relationship:(31)$$v_{\varphi p}=-v_{\mathrm{xp}}\tan\varphi$$

In addition, the compressive and tangential stresses acting on the material are also the same as that acting on the warhead surface.

This next section provides a concise and precise description of the experimental results, their interpretation, as well as the experimental conclusions that can be drawn.

## 4. Constitutive Equation of Concrete Material

In order to solve the equations more effectively, a state equation needs to be added. Concrete contains a large number of incompressible pebbles and pores, so it can still be regarded as a compressible material. The density change is caused by the normal compression of the non-granular material. From the perspective of volume compressibility, the ultimate density model can be adopted. In the free region, the average density of concrete $\rho =\rho_{0}$. In the response region, the material is subjected to the high-pressure impact compression, so the average density of concrete ρ is the ultimate density ρ. The Hugoniot curve is shown in Figure 4.

In terms of the velocity relationship on the boundary between the projectile and the target and the geometric equation of the warhead surface, for the entire response region, the tangential velocity of the target material in the response region is assumed to depend on the normal velocity by the relation $v_{\varphi }\left(x\right)=\beta_{\mathrm{tt}}v_{x}\left(x\right)\tan\varphi$, where $0\leq \beta_{\mathrm{tt}}\leq 1$ is the adhesion coefficient mentioned above. Then, for the stone particles with scale, the angular velocity can be approximated as
(32)$${\dot{\omega }}_{\theta }=\frac{1}{2}{\left(\nabla \times v\right)}_{\theta }=\frac{1}{2}\frac{v_{\varphi }}{l}$$

The tangential pressure divergence can be approximated as
(33)$${\left(\mathrm{div}\;p\right)}_{\varphi }=\frac{\partial p_{\varphi }}{\partial x}=\frac{p_{\varphi }}{l}$$

Equations (20)–(22) can be simplified as follows:(34)$$\frac{\partial \rho }{\partial t}+\frac{\partial \left({\rho v}_{x}\right)}{\partial x}+\frac{{\rho v}_{x}}{R+x}+\frac{{\rho v}_{x}}{\bar{R}+x}=0$$
(35)$$\{\begin{array}{c} \rho \frac{{\mathrm{Dv}}_{x}}{\mathrm{Dt}}=-\left(\frac{\partial p_{x}}{\partial x}+\frac{p_{x}}{R+x}+\frac{p_{x}}{\bar{R}+x}\right)\\ \rho_{s}\frac{{\mathrm{Dv}}_{\varphi }}{\mathrm{Dt}}=\frac{p_{\varphi }}{l} \end{array}$$
(36)$$\rho_{s}I_{\theta }\cdot \frac{1}{2}\frac{{\dot{v}}_{\varphi }}{l}=p_{\varphi }$$

By the state equation of the ultimate density, substituting $\rho_{\mathrm{sf}}=\rho^{*}$ into the discontinuity relationships, the following boundary conditions of the shock wave front can be obtained:(37)$$c_{x}=\frac{\rho^{*}}{\rho^{*}-\rho_{0}}v_{x}^{L}$$
(38)$$p_{x}^{L}=\rho^{*}\left(c_{x}-v_{x}^{L}\right)v_{x}^{L}=\rho_{0}c_{x}v_{x}^{L}$$
(39)$$p_{\varphi }^{L}=\rho^{*}\left(c_{x}-v_{x}^{L}\right)v_{\varphi }^{L}=\rho_{0}c_{x}v_{\varphi }^{L}$$
(40)$${\mathrm{lp}}_{\varphi }^{L}=\rho^{*}\left(c_{x}-v_{x}^{L}\right)I_{\theta }{\dot{\omega }}_{\theta }^{L}=\rho_{0}c_{x}I_{\theta }\frac{1}{2}\frac{v_{\varphi }^{L}}{l}$$

Substituting the relationship $\rho_{\mathrm{sf}}=\rho^{*}$ into the above basic equations and using the boundary conditions of the discontinuities above, the following solutions can be obtained separately:(41)$$v_{x}\left(x,t\right)=\left[\frac{R\bar{R}}{\left(R+x\right)\left(\bar{R}+x\right)}\right]v_{x}\left(0,t\right)$$
(42)$$p_{x}\left(0,t\right)=\rho^{*}\frac{\left(R\bar{R}\right)}{\left(R+x\right)\left(\bar{R}+x\right)}\left\{\left[\frac{\rho^{*}}{\rho^{*}-\rho_{0}}\frac{\left(R\bar{R}\right)}{\left(R+L\right)\left(\bar{R}+L\right)}-\frac{\left(R\bar{R}\right)}{\left(R+x\right)\left(\bar{R}+x\right)}\right]v_{x}^{2}\left(0,t\right)+\left(L-x\right){\dot{v}}_{x}\left(0,t\right)\right\}$$
(43)$$p_{\varphi }=\rho_{s}{l\dot{v}}_{\varphi }$$
where L is propagation distance of the shock wave. At the surface of the warhead (x=0), there are
(44)$$v_{x}\left(x,t\right)=v_{x}\left(0,t\right)$$
(45)$$p_{x}\left(0,t\right)=\rho^{*}\left\{\left[\frac{\rho^{*}}{\rho^{*}-\rho_{0}}\frac{\left(R\bar{R}\right)}{\left(R+L\right)\left(\bar{R}+L\right)}-1\right]v_{x}^{2}\left(0,t\right)+{L\dot{v}}_{x}\left(0,t\right)\right\}$$
(46)$$p_{\varphi }(0,t)=\rho_{s}{l\dot{v}}_{\varphi }\left(0,t\right)=-\rho_{s}{l\beta }_{\mathrm{tt}}{\dot{v}}_{x}\left(0,t\right)\tan\varphi$$

The total pressure of the warhead surface along the axial direction of the projectile is
(47)$$p_{z}(0,t)=p_{x}\left(0,t\right)\cos\varphi +p_{\varphi }(0,t)\sin\varphi$$

That is
(48)$$p_{z}(0,t)=\rho^{*}\left\{\left[\frac{\rho^{*}}{\rho^{*}-\rho_{0}}\frac{\left(R\bar{R}\right)}{\left(R+L\right)\left(\bar{R}+L\right)}-1\right]{\cos\varphi v}_{x}^{2}\left(0,t\right)+\left(\mathrm{Lcos}\varphi -\rho^{*}{l\beta }_{\mathrm{tt}}\tan\varphi \sin\varphi \right){\dot{v}}_{x}\left(0,t\right)\right\}$$

or (49)$$p_{z}(0,t)=\rho^{*}\left\{\left[\frac{\rho^{*}}{\rho^{*}-\rho_{0}}\frac{\left(R\bar{R}\right)}{\left(R+L\right)\left(\bar{R}+L\right)}-1\right]\cos^{3}{\varphi v}_{z}^{2}+\left({\mathrm{Lcos}}^{2}\varphi -\rho^{*}{l\beta }_{\mathrm{tt}}\sin^{2}\varphi \right){\dot{v}}_{z}\right\}$$

## 5. Dynamic Equations of the Projectile

When the projectile penetrates vertically into the concrete target containing stone particles, considering the projectile to be rigid, the dynamic equation is written by
(50)$$m_{p}{\dot{v}}_{z}=-\iint_{S_{A}}p_{z}(0,t)\mathrm{ds}$$

That is
(51)$$\left[m_{p}+m_{f}\left(t\right)\right]{\dot{v}}_{z}=-v_{z}^{2}\iint_{S_{A}\left(t\right)}\rho^{*}\left[\frac{\rho^{*}}{\rho^{*}-\rho_{0}}\frac{\left(R\bar{R}\right)}{\left(R+L\right)\left(\bar{R}+L\right)}-1\right]\cos^{3}\varphi \mathrm{ds}$$
where
(52)$$m_{f}\left(t\right)=\iint_{S_{A}\left(t\right)}\rho^{*}\left({\mathrm{Lcos}}^{2}\varphi -{l\beta }_{\mathrm{tt}}\sin^{2}\varphi \right)\mathrm{ds}$$

By means of the couple stress model, the calculation is carried out on the deceleration and velocity of a conical nose projectile as shown in Figure 5 perpendicularly penetrating against thick pebble-concrete target.

For a projectile with conical nose as shown in Figure 5, R=∞, $\frac{\left(\bar{R}\right)}{\left(\bar{R}+L\right)}=\frac{1}{1+\sin^{2}\varphi \left(\sqrt{\frac{\rho^{*}}{\rho^{*}-\rho_{0}}}-1\right)}$, $L=\mathrm{rsin}\varphi \left(\sqrt{\frac{\rho^{*}}{\rho^{*}-\rho_{0}}}-1\right)$, z=rtgφ. Therefore Equations (51) and (52) can be written by
(53)$$\{\begin{array}{c} \left[m_{p}+m_{f}\left(t\right)\right]{\dot{v}}_{z}=-v_{z}^{2}{\pi z}^{2}\frac{\cos^{5}\varphi }{\sin^{2}\varphi }\rho^{*}\left[\frac{\rho^{*}}{\rho^{*}-\rho_{0}}\frac{1}{1+\sin^{2}\varphi \left(\sqrt{\frac{\rho^{*}}{\rho *-\rho_{0}}}-1\right)}-1\right]z\leq h\\ \left[m_{p}+m_{f}\left(t\right)\right]{\dot{v}}_{z}=-v_{z}^{2}{\pi h}^{2}\frac{\cos^{5}\varphi }{\sin^{2}\varphi }\rho^{*}\left[\frac{\rho^{*}}{\rho^{*}-\rho_{0}}\frac{1}{1+\sin^{2}\varphi \left(\sqrt{\frac{\rho^{*}}{\rho^{*}-\rho_{0}}}-1\right)}-1\right]z\geq h \end{array}$$
(54)$$\{\begin{array}{c} m_{f}\left(t\right)={\pi \rho }^{*}z^{2}\left[\frac{2}{3}z\frac{\cos^{5}\varphi }{\sin^{2}\varphi }\left(\sqrt{\frac{\rho^{*}}{\rho^{*}-\rho_{0}}}-1\right)-{l\beta }_{\mathrm{tt}}\cos^{2}\varphi \right]z\leq h\\ m_{f}\left(t\right)={\pi \rho }^{*}h^{2}\left[\frac{2}{3}h\frac{\cos^{5}\varphi }{\sin^{2}\varphi }\left(\sqrt{\frac{\rho^{*}}{\rho^{*}-\rho_{0}}}-1\right)-{l\beta }_{\mathrm{tt}}\cos^{2}\varphi \right]z\geq h \end{array}$$

Equation (55) can be obtained from Equations (53) and (54) and the condition z=0, $v_{z}=v_{0}$.
(55)$$\{\begin{array}{c} {\dot{v}}_{z}=\frac{-v_{z}^{2}{\pi z}^{2}\rho^{*}}{\left[m_{p}+m_{f}\left(t\right)\right]}\frac{\cos^{5}\varphi }{\sin^{2}\varphi }\left[\frac{\rho^{*}}{\rho^{*}-\rho_{0}}\frac{1}{1+\sin^{2}\varphi \left(\sqrt{\frac{\rho^{*}}{\rho^{*}-\rho_{0}}}-1\right)}-1\right]z\leq h\\ {\dot{v}}_{z}=\frac{-v_{z}^{2}{\pi h}^{2}\rho^{*}}{\left[m_{p}+m_{f}\left(t\right)\right]}\frac{\cos^{5}\varphi }{\sin^{2}\varphi }\left[\frac{\rho^{*}}{\rho^{*}-\rho_{0}}\frac{1}{1+\sin^{2}\varphi \left(\sqrt{\frac{\rho^{*}}{\rho^{*}-\rho_{0}}}-1\right)}-1\right]z\geq h \end{array}$$

## 6. Comparison of the Calculated Examples

This paper focuses on the mechanical characteristics of high-speed impact. Therefore, in this section, when the projectile velocity is in the high-speed zon (v≥800 m/s), the theoretical calculation and the FE simulation of deceleration and velocity are compared.

### 6.1. Theoretical Calculation

Based on the dynamic equations of the projectile in Section 5, the following conditions are brought in, and the calculation have been performed by numerical calculation software. This paper considers that the initial density of concrete ρ is $2440\mathrm{kg}/m^{3}$ and the ultimate density $\rho^{*}$ is $2640\mathrm{kg}/m^{3}$. Other conditions are as follows: $m_{p}=3\;\mathrm{kg}$, h=0.1 m, d/2=0.03 m, $v_{0}=1000m/s$, $\mathrm{tg}\varphi =2\;h/d\;\mathrm{and}\;\beta_{\mathrm{tt}}=0.5$ F or different scales of pebbles (thelongaxislengthl=0,0.03 m∧0.06 m), when the projectile velocity is still in the high-speed zone (v≥800 m/s), the curves of deceleration of the projectile are calculated and shown in Figure 6. When there is no pebble in the concrete material, the peak value of deceleration is about 54,000 g. When the scale of pebbles is 0.06 m, the peak value of deceleration is about 71,000 g. The corresponding velocity curves are shown in Figure 7.

### 6.2. The FE Simulation

This paper uses random ellipsoid to simulate the aggregate. According to LV [2], on the basis of ensuring the validity of the simulation, the idea of establishing the 3D meso-scale model and the meshing size can meet the needs of the comparative theoretical analysis. At the same time, the long axis length of the ellipsoidal pebble is controlled. Figure 8 shows the geometric dimensions of the target model with the aggregate (l=0.06 m) and that of other target models (l=0, 0.03 m) are the same. The geometric dimensions of the projectile are shown in Figure 9.

According to the hypothesis proposed in this paper, the projectile is regarded as a rigid body, so MAT_RIGID is adopted for the projectile. Through the theoretical simulation and experiments, it is found that when considering the meso-scale model of concrete, the model parameters of MAT_JOHNSON_HOLMQUIST_CONCRETE provided by Holmquist cannot effectively simulate the actual situation of concrete [7,8,9]. Therefore, when using a concrete meso-scale model for the FE simulation, it is necessary to re-determine the material parameters of mortar and pebbles. In this section, the material parameters in Table 1, Table 2, Table 3 and Table 4 are determined through repeated simulation tests in combination with the material parameters of [2,23].

It should be noted that in the FE simulation, the rotation movement of pebbles cannot be intuitively expressed. In the FE simulation, by extracting the displacement of any four points (point A–point D) on the pebble aggregate, it is found that the displacement curves of the four points are different, as shown in Figure 10. Combined with the experimental phenomenon at the beginning of this paper, the pebbles can ensure relative integrity in the process of penetration. It shows that the pebbles do not only perform translational motion during the penetration. To some extent, it can be considered that the splashed pebbles have rotational motion.

The simulation results still focus on the variation of deceleration and velocity with time. The dotted lines in Figure 11 are the deceleration curves of the projectile penetrating different sizes of aggregates in the FE simulation. If there is no pebble in concrete material, the peak value of deceleration is about 51,000 g. If the scale of pebbles is 0.06 m, the peak value of deceleration is about 74,000 g. The dotted lines in Figure 12 are the corresponding velocity curves.

It can be seen from Figure 11 and Figure 12 that the consistency between theoretical calculation and simulation results is good. From the deceleration results, it can be observed that the pebbles will directly affect the deceleration of the projectile. If the pebble scales continue to increase, the peak value of deceleration will continue to increase.

## 7. Conclusions

For most concrete targets, there are several pebbles to strengthen the structures. The method is proposed in this paper for analyzing the effects of pebbles on the resistant forces of targets on the projectile. Firstly, an orthogonal curvilinear coordinate system is constructed, which effectively describes and perfects the normal cavity expansion theory. Then, based on this theory, a couple stress model is proposed using the basic equations and discontinuity theory of continuum mechanics. The model describes the effects of the translational and rotational motion of pebble aggregates on the resistant force applied on the warhead during the penetration. The theoretical analysis is verified by the simulation of the 3D meso-scale model. Numerical calculations and simulation results show that, with the increasing of the pebble scales, the peak values of deceleration will increase. This means that the larger the scale of stone particles, the stronger the effect of movement of the pebble on the resistant force is applied on the warhead.

## Figures and Tables

**Figure 1 materials-15-01675-f001:**
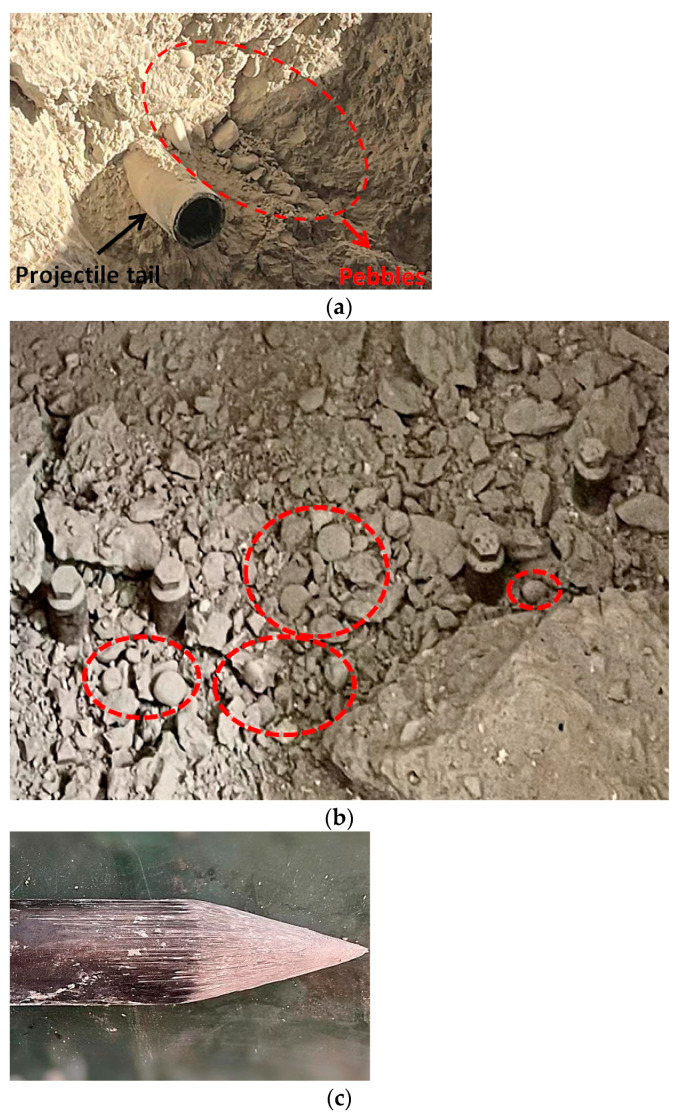
Correlation penetration experiment [6]. (**a**) The pebble aggregate in the crater. (**b**) Obvious pebbles splashed out. (**c**) The warhead surface after the penetration experiment.

**Figure 2 materials-15-01675-f002:**
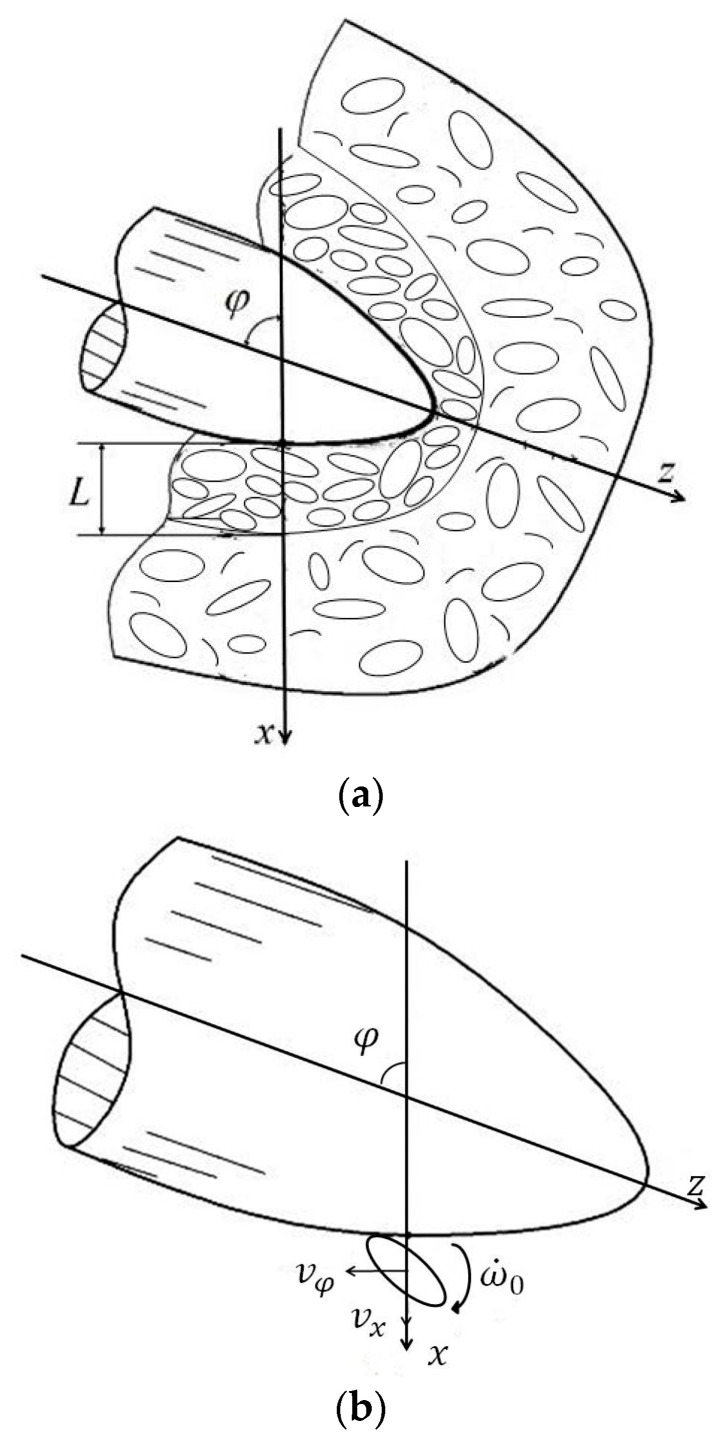
The penetration of a projectile against the pebble-concrete target. (**a**) Two regions of the target medium. (**b**) The effect of particles on the surface of the warhead.

**Figure 3 materials-15-01675-f003:**
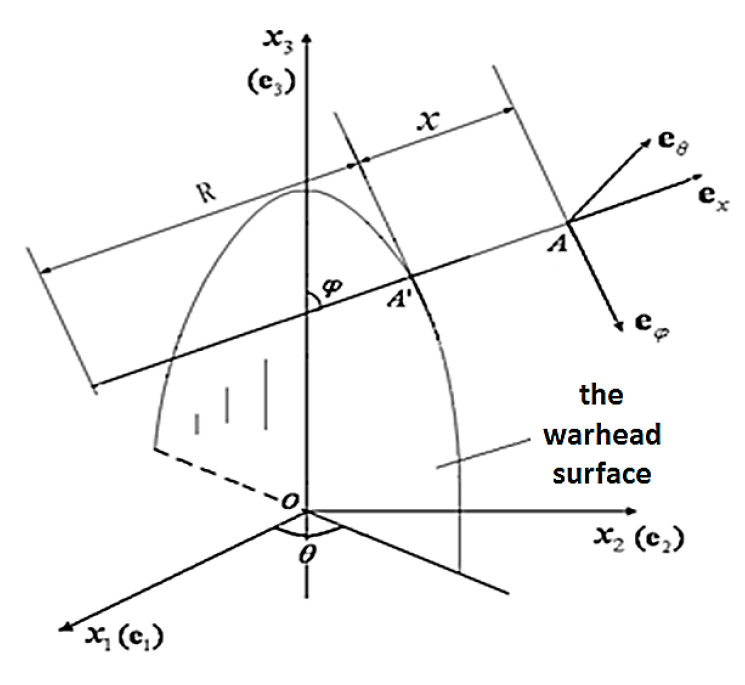
Orthogonal curvilinear coordinate system.

**Figure 4 materials-15-01675-f004:**
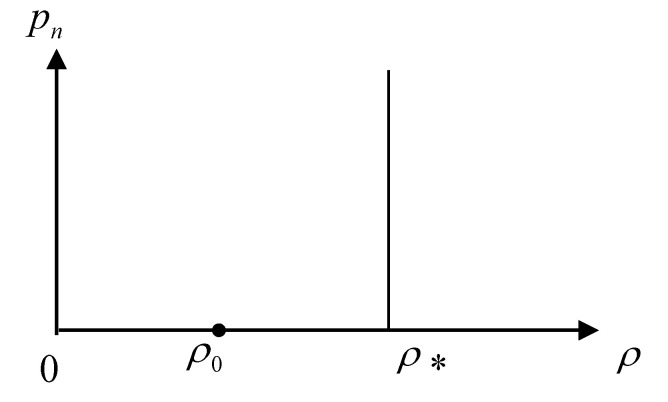
The Hugoniot curve of concrete.

**Figure 5 materials-15-01675-f005:**
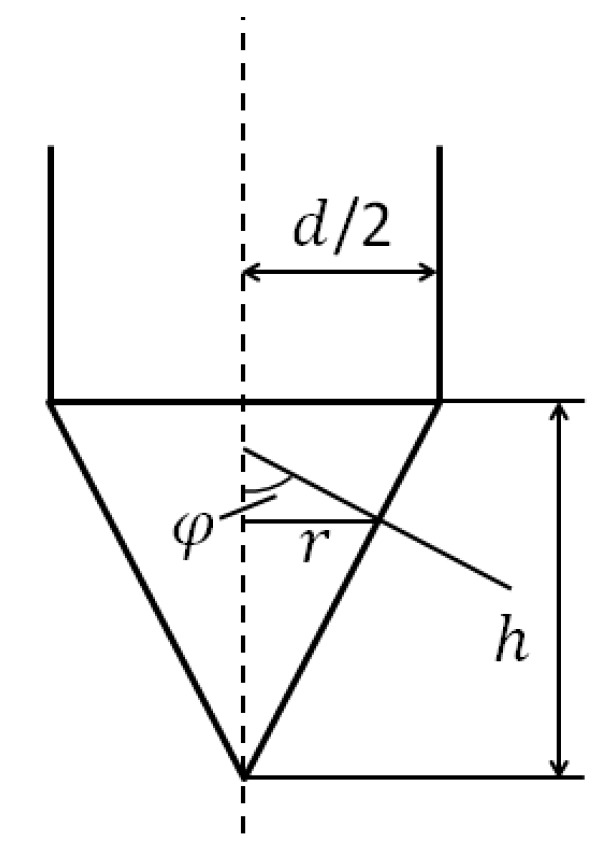
The projectile nose used.

**Figure 6 materials-15-01675-f006:**
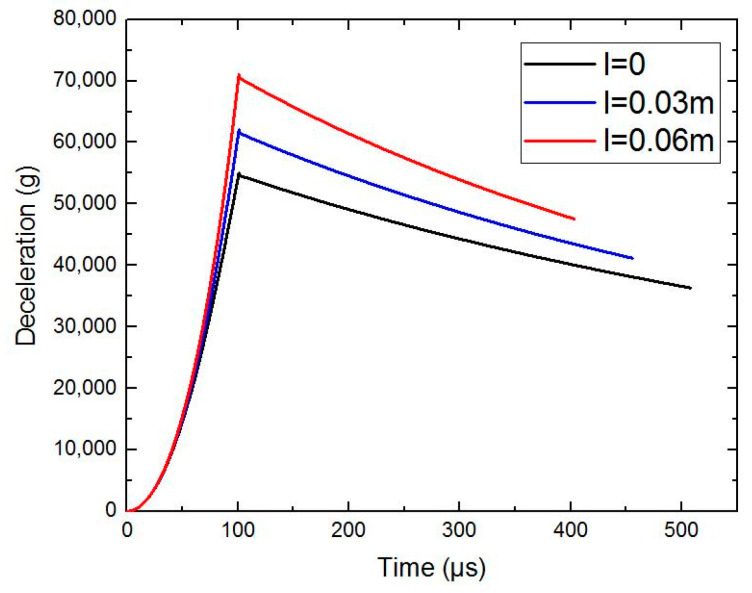
The deceleration curves of the theoretical calculation.

**Figure 7 materials-15-01675-f007:**
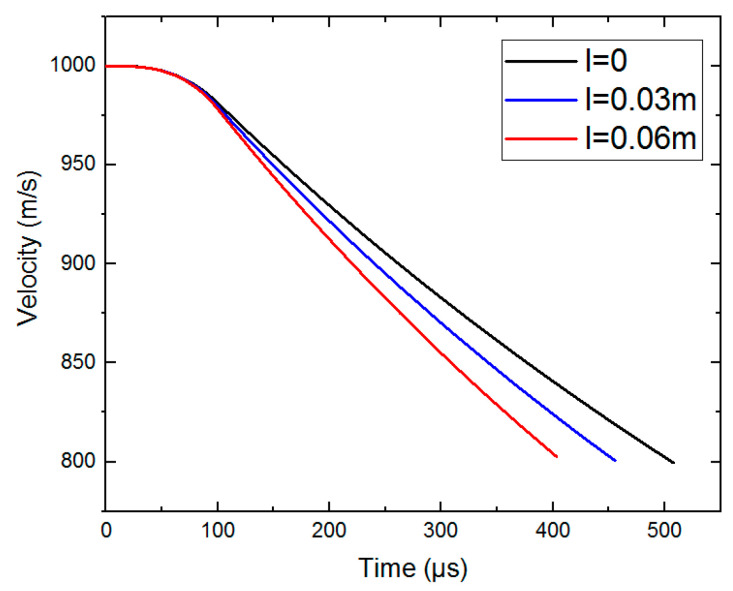
The velocity curves of the theoretical calculation.

**Figure 8 materials-15-01675-f008:**
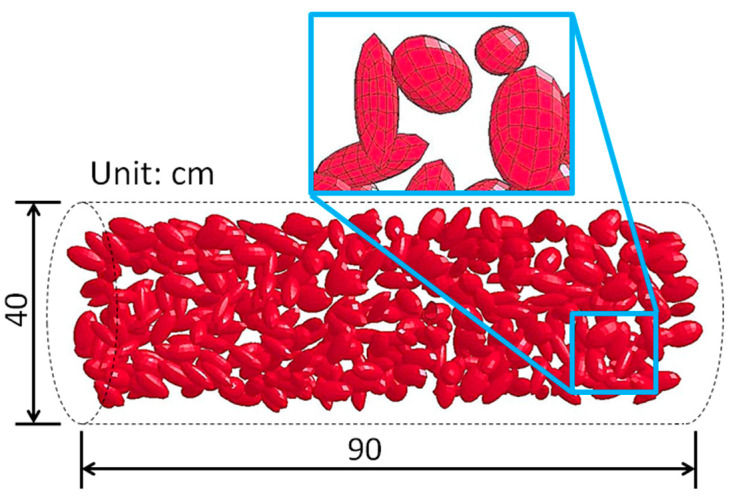
The geometric dimension of the target containing aggregate (l = 0.06 m).

**Figure 9 materials-15-01675-f009:**
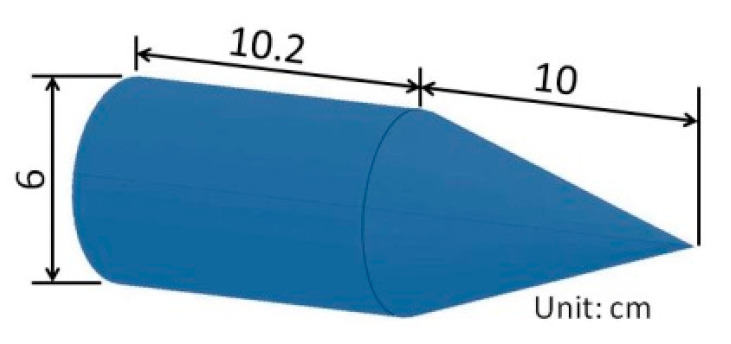
The geometric dimensions of the projectile.

**Figure 10 materials-15-01675-f010:**
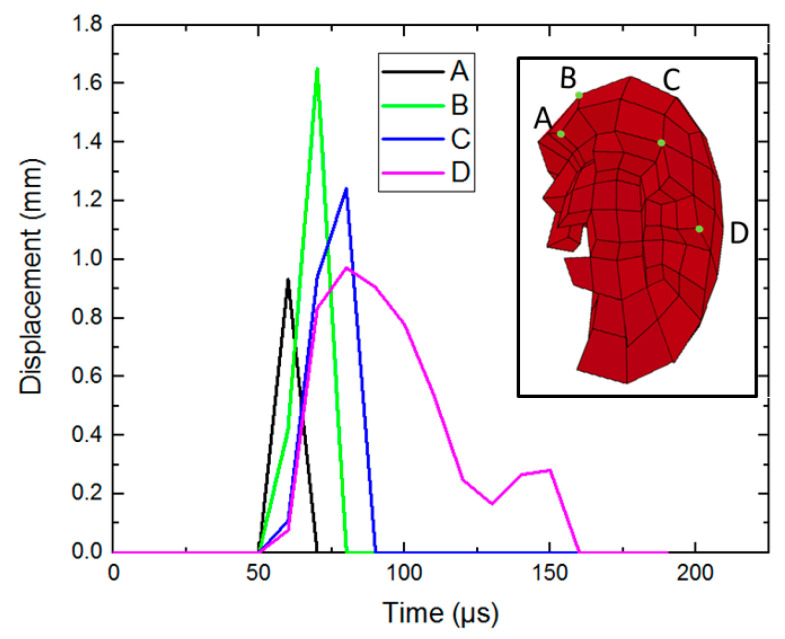
The displacement of the different nodes of the penetrated pebble.

**Figure 11 materials-15-01675-f011:**
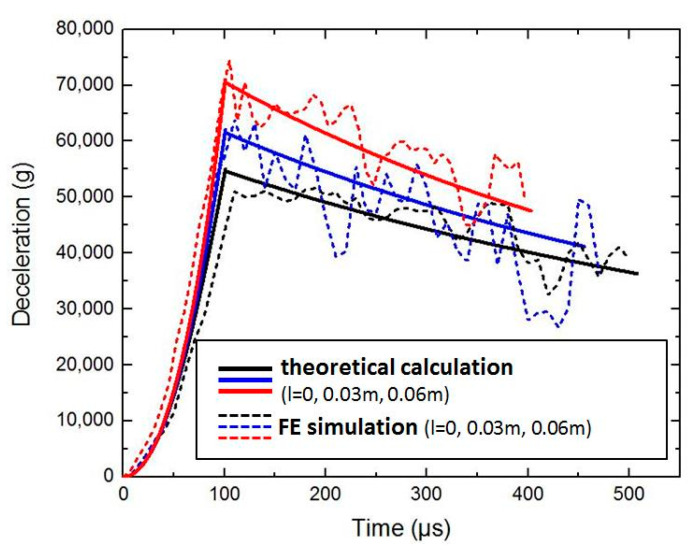
The deceleration curves of the FE simulation.

**Figure 12 materials-15-01675-f012:**
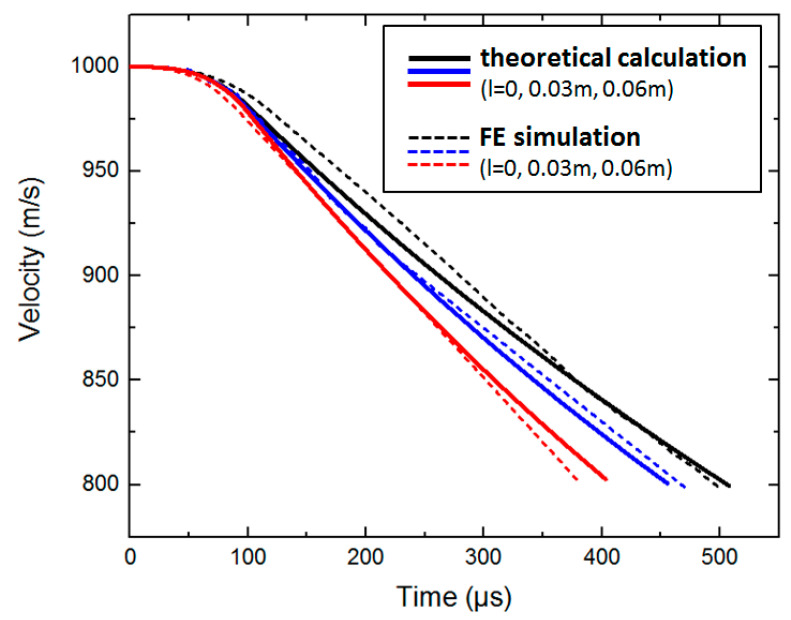
The velocity curves of the FE simulation.

**Table 1 materials-15-01675-t001:** Model coefficients of the projectile (unit system: cm − g − μs).

Density ρ	Young’s Modulus (ρ)	Poisson’s Ratio (PR)
7.85	300	0.269

**Table 2 materials-15-01675-t002:** Model coefficients of concrete (unit system: cm − g − μs) [24].

Density ρ	Shear Modulus (G)	Normalized Cohesive Strength (A)	Normalized Pressure Hardening (B)	Strain Rate Coefficient (C)	Pressure Hardening Exponent (N)	$\mathrm{Quasi}\text{-}\mathrm{Static}\;\mathrm{Uniaxial}\;\mathrm{Compressive}\;\mathrm{Strength}\;f_{c}$
2.44	0.1486	0.79	1.6	0.007	0.61	0.00048
**Maximum Tensile Hydrostatic Pressure (** **T** **)**	**Reference Strain Rate (** **EPSO** **)**	**Amount of Plastic Strain before Fracture (** **EFMIN** **)**	**Normalized Maximum Strength (** **SFMAX** **)**	**Crushing Pressure** $P_{crush}$	**Crushing Volumetric Strain** $\mu_{crush}$	**Locking Pressure** $P_{lock}$
0.00004	1.0	0.01	7	0.00016	0.001	0.008
**Locking Volumetric Strain** $\mu_{lock}$	**Damage Constant** $D_{1}$	**Damage Constant** $D_{2}$	**Pressure Constant** $K_{1}$	**Pressure Constant** $K_{2}$	**Pressure Constant** $K_{3}$	
0.1	0.04	1.0	0.85	−1.71	2.08	

**Table 3 materials-15-01675-t003:** Model coefficients of concrete mortar (unit system: cm − g – μs) [2,24].

Density ρ	Shear Modulus (G)	Normalized Cohesive Strength (A)	Normalized Pressure Hardening (B)	Strain Rate Coefficient (C)	Pressure Hardening Exponent (N)	$\mathrm{Quasi}\text{-}\mathrm{Static}\;\mathrm{Uniaxial}\;\mathrm{Compressive}\;\mathrm{Strength}\;f_{c}$
2.2	0.12	0.8	1.8	0.1	0.65	0.00038
**Maximum Tensile Hydrostatic Pressure (** **T** **)**	**Reference Strain Rate (** **EPSO** **)**	**Amount of Plastic Strain before Fracture (** **EFMIN** **)**	**Normalized Maximum Strength (** **SFMAX** **)**	**Crushing Pressure** $P_{crush}$	**Crushing Volumetric Strain** $\mu_{crush}$	**Locking Pressure** $P_{lock}$
0.00003	1.0	0.003	4	0.000167	0.001	0.002
**Locking Volumetric Strain** $\mu_{lock}$	**Damage Constant** $D_{1}$	**Damage Constant** $D_{2}$	**Pressure Constant** $K_{1}$	**Pressure Constant** $K_{2}$	**Pressure Constant** $K_{3}$	
0.01	0.06	1.0	0.85	−1.71	2.08	

**Table 4 materials-15-01675-t004:** Model coefficients of concrete aggregate (unit system: cm − g − μs) [2,24].

Density ρ	Shear Modulus (G)	Normalized Cohesive Strength (A)	Normalized Pressure Hardening (B)	Strain Rate Coefficient (C)	Pressure Hardening Exponent (N)	$\mathrm{Quasi}\text{-}\mathrm{Static}\;\mathrm{Uniaxial}\;\mathrm{Compressive}\;\mathrm{Strength}\;f_{c}$
2.66	0.215	0.9	2	0.1	0.65	0.0016
**Maximum Tensile Hydrostatic Pressure (** **T** **)**	**Reference Strain Rate (** **EPSO** **)**	**Amount of Plastic Strain before Fracture (** **EFMIN** **)**	**Normalized Maximum Strength (** **SFMAX** **)**	**Crushing Pressure** $P_{crush}$	**Crushing Volumetric Strain** $\mu_{crush}$	**Locking Pressure** $P_{lock}$
0.0001	1.0	0.02	4	0.00053	0.0012	0.008
**Locking Volumetric Strain** $\mu_{lock}$	**Damage Constant** $D_{1}$	**Damage Constant** $D_{2}$	**Pressure Constant** $K_{1}$	**Pressure Constant** $K_{2}$	**Pressure Constant** $K_{3}$	
0.01	0.08	1.0	1.4	−20	25	

## Data Availability

Not applicable.

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
