# Peer review of "Dynamics Analysis of Penetration against Pebble-Concrete Target"

_materials, 2022, doi:10.3390/ma15051675_

Round 1
Reviewer 1 Report
Author investigated Dynamics Analysis of Penetration against Pebbles-Concrete Target
- Introduction first full paragraph is without reference and also the Figures are without reference. The storyline could be more robust.
- Novelty part is missing or lost in your writing? why did you perform this work? I hope you are doing this work to let readers know about the available mathematical theories.
- section 6.3- comparison between theoretical calculation and simulation" how these are different?
- Figure 12: how to read point A B C in the insect figure
- Please check the jump from equation 5 to 6
- Please explain again the Figure 10 why the transient behaviour till 100 and then semi transient nature
- can we use any modern techniques to solve these issues while not using any traditional classical theories?https://doi.org/10.3390/en14092404 such as deep learning?
Reviewer 2 Report
The authors proposed the employment of the couple stress theory based on the normal cavity expansion for the effect of pebbles on projectile. The proposed method is compared with the finite element method simulation, and agreeement between two approaches was shown. It was also demonstrated that the larger the scale of pebble particles, the stronger the effect of rotation on the resistant force applied on the warhead.
The paper is recommended for publication after some revisions:
1). The only example of the projectile of specific shape was shown. What is the accuracy of the proposed method for other shapes?
2). The reviewer cannot fully agree that "that the consistency between theoretical 340 calculation and simulation results is good" in Figure 11. Indeed, the behaviour is predicted well qualitatively, but there is a certain misalignment for the curves predicted by the presented method and by the FEM for t>50 mus. Could you please explain the specific discrepancy?
3) Editing of English language is also required.
4) Several values given in Tables 2-4 are not described accurately, e.g. SFMAX, EFMIN etc.
Round 2
Reviewer 1 Report
all comments are addressed